# WHY DO NEURAL RESPONSE GENERATION MODELS PREFER UNIVERSAL REPLIES?

## ABSTRACT

Recent advances in neural Sequence-to-Sequence (Seq2Seq) models reveal a purely data-driven approach to the response generation task. Despite its diverse variants and applications, the existing Seq2Seq models are prone to producing short and generic replies, which blocks such neural network architectures from being utilized in practical open-domain response generation tasks. In this research, we analyze this critical issue from the perspective of the optimization goal of models and the specific characteristics of human-to-human conversational corpora. Our analysis is conducted by decomposing the goal of Neural Response Generation (NRG) into the optimizations of word selection and ordering. It can be derived from the decomposing that Seq2Seq based NRG models naturally tend to select common words to compose responses, and ignore the semantic of queries in word ordering. On the basis of the analysis, we propose a max-marginal ranking regularization term to avoid Seq2Seq models from producing the generic and uninformative responses. The empirical experiments on benchmarks with several metrics have validated our analysis and proposed methodology.

## 1 INTRODUCTION

Past years have witnessed the dramatic progress on the application of generative sequential models (also noted as seq2seq learning (Sutskever et al., 2014; Bahdanau et al., 2015)) on Neural Response Generation (NRG) fields (Vinyals & Le, 2015; Serban et al., 2017). Seq2seq model has been proved to be capable of directly generating reply given an open domain query (Li et al., 2016c; Xing et al., 2017). Both relevant words or phrases are automatically selected, and smoothness and fluency of responses are guaranteed through the end-to-end learning. Moreover, abundant impressive human-to-machine conversation cases have been presented in many previous studies (Serban et al., 2016; Shang et al., 2015; Shao et al., 2017).

Despite these promising results, current Sequence-to-Sequence (Seq2Seq) architectures for response generation are still far from steadily generating relevant and coherent replies. The essential issue identified by many studies is the *Universal Replies*: the model tends to generate short and general replies which contain limited information, such as "That's great!", "I don't know", etc. (Li et al., 2016b;d; Mou et al., 2016; Xing et al., 2017). Intuitively, this problem was attributed to the vast coverage of common replies in the training set and insufficient guiding knowledge in the models' response generation step (Mou et al., 2016; Shao et al., 2017). Hence, current efforts mainly focus on introducing external information to the model (Mou et al., 2016; Xing et al., 2017), and encouraging the model to generate diverse responses in searching space via variational beam search strategies during inference (Shao et al., 2017; Li et al., 2016b;d).

Nevertheless, most previous analysis over the issue are empirical and lack of statistical evidence. Therefore, in this paper, we conduct an in-depth investigation on the performance of seq2seq models on the NRG task. In our inspections on the existing dialog corpora, it is shown that those repeatedly appeared replies have two essential traits: 1) Most of them are composed of highly frequent words; 2) They cover a large portion of the dialog corpora that each universal reply stands for the response of various queries. Above characteristics of universal replies deviate the NRG from other successful applications of sea2seq model such as translation, and lead current generative NRG models to prefer common replies. To discuss the influences from the specific distributed corpus, we decompose the target sequence's probability into two parts and analyze the probability respectively.

Table 1: Replies and translated version of an example which reveal the different source-target sentence distribution for dialog and translation.

| Query | I would add Metropolis to the list. |
|---|---|
| Replies | I love this film so much. 
 Me too, it is a beautiful film. 
 This movie has beautiful background art. 
 Fritz is really a good director, I like his film. 
 Is "Metropolis" based on a book? 
 Brigitte cooling off on the set of Metropolis. |
| Translate | J'ajouterais Metropolis à la liste. 
 Je voudrais ajouter Metropolis à la liste. |

To break down the mentioned characteristics of dialog corpora in the model training step, we propose a ranking-oriented regularization term to prune the scores of those irrelevant replies. Experimental results reveal that the model with such regularization can produce better results and avoid generating ambiguous responses. Also, case studies show that the issue of generic response is alleviated that these common responses are ranked relatively lower than more appropriate answers.

The main contributions of this paper are concluded as follows: 1) We analyze the loss function of Seq2seq models on NRG task and conclude several critical reasons that the NRG models prefer universal replies; 2) Based on the analysis, a max-marginal ranking regularization is presented to help the model converge to informative responses.

## 2 ANALYSIS OF SEQ2SEQ MODELS FOR NRG

Different from significant advances in machine translation (Bahdanau et al., 2015) and abstractive summarization (Rush et al., 2015; Nallapati et al., 2016), it remains challenging to apply Seq2Seq models in practical response generation. One widely accepted issue within current models is that Seq2Seq architectures are inclined to produce common and unrelated replies, even when the quality of training data is significantly improved and different Seq2Seq variants are proposed. The primary reason for this phenomenon lies in the fact that the semantic constraint from query to the possible responses is naturally weak, since the responses to a given query are not required to be semantically equivalent. In contrast, the references in machine translation or summarization are usually restricted to be equivalent to each other semantically or even lexically. Especially, for machine translation, words that appear in the target language should satisfy word level mapping from the source sentence, so the learned word alignment function could ensure the model to generate suitable translated words. Different from learning the semantic alignments between languages in NMT, in NRG the replies can be diversified as they only need to satisfy the causality with the given queries. Moreover, given a query, the sequential model is optimized to learn the shared information among all replies, thus the model is more likely to choose those high-frequent common replies, which is also mentioned in Ritter et al. (2011).

Taking the case in Table 1 for example, the topic of this query is about movie. It can be observed that the replies shown in the table are semantically diversified: the first two replies are related to the opinion of the respondent toward the movie, while the rest of the replies are about the director, content, and origin of the movie. By contrast, the two valid translations in French are very similar regarding their semantics, which can be attributed to the fixed word-level mapping between query and targets.

### 2.1 PROBLEM DECOMPOSITION

The sequence-mapping problem in NRG can be decomposed into two independent sub-learning problems: 1) *Target word selection*, in which a query is summarized and translated into the semantic space of responses, and then a set of target words is selected to represent the meaning; 2) *Word ordering*, in which a grammatical coherent reply is generated based on the candidate word set (Vinyals et al., 2016). The word selection and ordering of the target sequence are jointly learned which can

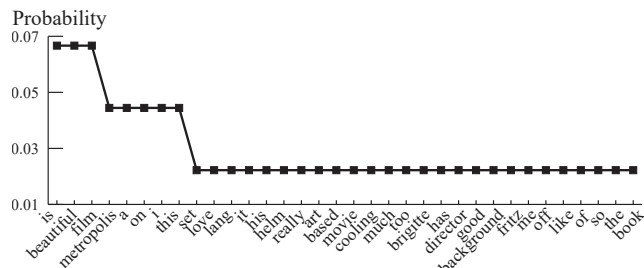

Figure 1: Response Unigram probability distribution in Table 1.

also be reflected in the model's loss function by two possible factored phases:

$$-\log p(y|x) = -\log p(\mathcal{S}(y)|x) - \log p(y|\mathcal{S}(y), x) \tag{1}$$

where $x$ stands for the given query and $y$ is the corresponding response with $n$ words. Besides, $\mathcal{S}(y) = \{w_1, \cdots, w_n | w_i \in y, i \in [1, n]\}$ represents all predicted words without sequential order, so $p(\mathcal{S}(y)|x)$ is referred as the probability of the *target word selection*. Meanwhile, $p(y|\mathcal{S}(y), x)$ indicates the probability of *word ordering* given this group of possible words. Thus, the objective can be redescribed from maximizing the probability of the ground truth response $y$ under query $x$ to maximizing these two joint probabilities simultaneously.

After the above interpretation, we will further discuss the impact of the implicative constriction from two separated probabilities in Eq. 1, which results in the potential failure of models in learning conversational patterns.

## 2.2 TARGET WORD SELECTION PROBABILITY

Assuming that we have a set of $\mathcal{K}$ ground-truth replies: $\{y_1, \cdots, y_\mathcal{K}\}$ to a given query $x$, the upper bound of the *target word selection* probability can be derived via Jensen's Inequality (Boyd & Vandenberghe, 2004):

$$
\begin{aligned}
\sum_k^\mathcal{K} \log p(\mathcal{S}(y_k)|x) &= \sum_k^\mathcal{K} \log \prod_{w \in \mathcal{S}(y_k)} p(w|x) \\
&= \sum_{w \in \cup_k^\mathcal{K} \mathcal{S}(y_k)} \log p(w|x) \\
&\leq L_\mathcal{S} \log \sum_{w \in \cup_k^\mathcal{K} \mathcal{S}(y_k)} \frac{p(w|x)}{L_\mathcal{S}}
\end{aligned}
\tag{2}
$$

where $\cup_k^\mathcal{K} \mathcal{S}(y_k)$ denotes all the words appearing in the entire response set, and $L_\mathcal{S} = |\cup_k^\mathcal{K} \mathcal{S}(y_k)|$. Thus, optimizing the first segment is proportional to maximizing the last conditional probabilities, and the optimal strategy is to assign probabilities according to the frequency of words in these $\mathcal{K}$ responses. Such strategy adopted by Seq2Seq can be verified by the long-tailed distribution of words in Fig. 1, in which only few common words are assigned with preferred high probabilities. Given that, during the inference, the best strategy is to employ more frequently occurring words rather than rare ones such as "background," "art," and "director" in Table 1.

Furthermore, assuming that each response contains a fixed number of $T$ words (so that $1 \leq L_\mathcal{S} \leq \mathcal{K} \times T$), we can find that the probability of each response for $x$ is inversely proportional to $\mathcal{K}$:

$$L_\mathcal{S} \log \sum_{w \in \cup_k^\mathcal{K} \mathcal{S}(y_k)} \frac{p(w|x)}{L_\mathcal{S}} = L_\mathcal{S} \log \frac{\mathbb{E}(w|x) \times T}{\mathcal{K} \times T \times L_\mathcal{S}} \propto \log \frac{1}{(\mathcal{K} \times L_\mathcal{S})^{L_\mathcal{S}}} \leq \log \frac{1}{\mathcal{K}} \tag{3}$$

where $\mathbb{E}(w|x)$ denotes the mean frequency of words appeared in these $\mathcal{K}$ replies, which is 1.32 for the cases in Table 1. In general, the mean frequency is around 1 owing to the long-tailed Unigram distribution which satisfies Zipf's law (Zipf, 1935). In other words, the *target word selection*

*probability* is limited by $\mathcal{K}$, so queries with more diverse answers are more challenging to learn. Meanwhile, it is difficult to obtain good predictions for lower-informational queries, as they contain more possible responses which are somewhat equivalent to a larger $\mathcal{K}$ (Li et al., 2016a).

Nonetheless, the translation task requires word-level mappings as they are well-aligned in the semantic space, therefore source and target sentences are semantically equivalent. So that, translated candidates are confined to $\mathcal{K} \approx 1$. Thus the upper bound can be approximated as the full probability.

## 2.3 WORD ORDERING PROBABILITY

### 2.3.1 LEMMAS

Before discussing the *word ordering probability*, we present four lemmas and corresponding proofs. Moreover, all these lemmas are only available for the response generation task except Lemma 1.

According to the Zipf's law (Zipf, 1935), the frequency of any word is inversely proportional to its rank in the frequency table, such that the probability $p(w_i) = Z/i^\alpha$, where $Z \approx 0.1$, $\alpha \approx 1$, and $i$ is the frequency rank of the word $w_i$. Then, denoting the vocabulary size as $V$ and the total number of query-response pairs as $N$, we can formulate two characteristics of a universal reply $y$ as follows:

1) A response is universal if it consists of only top-$t$ ranked words. For any word $w$ in such response, $p(w) \geq 1/(10t)$ according to the Zipf's law.

2) The amount of possible queries $M$ of $y$ is directly proportional to the size of query-response pairs $N$, noted as $1 \ll M \propto N$.

To simplify, we suppose that $t > 1000$ to cover most universal replies, and the frequency of the response not belonging to the universal replies is a constant $c$ ($1 \leq c \ll M$). Accordingly, we can derive the following lemmas.

**Lemma 1** $p(\mathcal{S}(y)|y) = 1$, $p(\mathcal{S}(y), y) = p(y)$, $p(x, y, \mathcal{S}(y)) = p(x, y)$.

*Proof.* Lemma 1 describes the obvious fact that the event "the word set of the response equals to $\mathcal{S}(y)$" must happen when the event "$y$ stands for the response" is established.

**Lemma 2** $p(x|y_{ur}) = \epsilon_1$, *where $\epsilon_1 > 0$ and is sufficiently small, and $y_{ur}$ is a universal reply.*

*Proof.* Based on the second character of the universal reply and the fact that $N$ is a very large number for any large scaled datasets, Lemma 2 is established as: $p(x|y_{ur}) = \frac{1}{M} \propto \frac{1}{N} = \epsilon_1$

**Lemma 3** $\sum_i p(y_i^{ur}|\mathcal{S}(y)) \to 1$, $p(y_j^o|\mathcal{S}(y)) = \epsilon_2$, *where $\epsilon_2 > 0$ and is sufficiently small, $y_i^{ur}$ stands for the $i$-th universal reply and $y_j^o$ is the $j$-th non-universal grammatical replies, meanwhile, $\mathcal{S}(y_i^{ur}) \subseteq \mathcal{S}(y)$ and $\mathcal{S}(y_j^o) \subseteq \mathcal{S}(y)$*

*Proof.* According to the following inequation $\sum_i^t \frac{1}{i} > \int_1^{t+1} \frac{1}{x} dx = ln(t+1)$, we can get the conclusion that the probability of a chosen word belonging to the most frequent $t$ words is large than $0.1 * ln(t+1) > 0.69$. Since $y$ contains $T$ words, there is at least $Tln(t+1)$ words belonging to the top-$t$ ranked on average according to the binomial distribution.

We suppose $m$ responses are universal replies among the $n$ possible responses when their words are constrained by $\mathcal{S}(y)$. Besides, the proportion of m can be computed as:

$$
\begin{aligned}
\frac{m}{n} &= \sum_{i=1}^{Tln(t+1)} \frac{C_T^i}{\sum_{j=1}^{T} C_T^j} * \frac{1}{10}ln(t+1) \\
&= \frac{2^T - \sum_{i=Tln(t+1)}^{T} C_T^i}{2^T} * \frac{1}{10}ln(t+1) \qquad (4) \\
&> \frac{1}{20}ln(t+1) \\
&> 0.34
\end{aligned}
$$

where $C$ donates the combination. Since $n/m$ is not a very large number, the total probability of these $m$ replies can be deducted as:

$$
\begin{aligned}
\sum_i p(y_i^{ur}|\mathcal{S}(y)) &= \frac{\sum_i^m f(y_i^{ur})}{\sum_i^m f(y_i^{ur}) + \sum_i^{n-m} f(Y_i^o)} \\
&= \frac{M * m}{M * m + c * (n - m)} \\
&= \frac{M}{M + n/m - c} \\
&> \frac{M}{M + 3 - c}
\end{aligned}
\tag{5}
$$

where $f(y)$ donates the frequency of a response $y$ in the corpus. According to the Eq. 5 and the fact that $M \propto N$ is a very large number for any practical large-scale datasets, $\sum_i p(y_i^{ur}|\mathcal{S}(y)) \to 1$ can be established. Apparently, for any other candidate response $y_j^o$, its probability satisfies $p(y_j^o|\mathcal{S}(y)) < 1 - \sum_i p(y_i^{ur}|\mathcal{S}(y)) = \epsilon_2$.

**Lemma 4** *Assuming each informative query has $\mathcal{K}$ ground-truth replies and the query-response pairs are extracted from a multi-turn conversational corpus, a reply $y$ not belonging to universal replies has $\mathcal{K}$ unique queries, noted as $p(x|y) = \frac{1}{\mathcal{K}}$.*

*Proof.* Most query-response pairs are extracted from a practical large-scale multi-turn conversational corpus, so that any response always works as the post in another pair. That is, $y$ also appears $\mathcal{K}$ times as it also has $\mathcal{K}$ replies. Therefore, there also exist $\mathcal{K}$ unique posts for $y$.

### 2.3.2 DISCUSSION

On the basis of Lemma 1, the *word ordering probability* could be deducted as:

$$
\begin{aligned}
log\, p(y|\mathcal{S}(y), x) &= log\frac{p(\mathcal{S}(y)|y)\, p(y)\, p(x|y, \mathcal{S}(y))}{p(\mathcal{S}(y))\, p(x|\mathcal{S}(y))} \\
&= log1 + log\frac{p(y)}{p(\mathcal{S}(y))} + log\frac{p(x|y, \mathcal{S}(y))}{p(x|\mathcal{S}(y))} \\
&= log\frac{p(y, \mathcal{S}(y))}{p(\mathcal{S}(y))} + log\frac{p(x, y, \mathcal{S}(y))\, p(\mathcal{S}(y))}{p(y, \mathcal{S}(y))\, p(x, \mathcal{S}(y))} \\
&= log\, p(y|\mathcal{S}(y)) + log\frac{p(x, y)\, p(\mathcal{S}(y))}{p(y)\, p(x, \mathcal{S}(y))} \\
&= log\, p(y|\mathcal{S}(y)) + log\frac{p(x|y)}{p(x|\mathcal{S}(y))} \\
&= log\, p(y|\mathcal{S}(y)) + log\frac{p(x|y)}{\sum_i p(x|y_i)p(y_i|\mathcal{S}(y))}
\end{aligned}
\tag{6}
$$

All the possible $y_i$ satisfying $\mathcal{S}(y_i) \subseteq \mathcal{S}(y)$ can be divided into three categories: ground-truth reply $y$, universal replies $y^{ur}$ and other replies $y^o$. From above, we can get the following direct proportion according to the Lemma 2 and Lemma 3,

$$
\begin{aligned}
&\sum_i p(x|y_i)p(y_i|\mathcal{S}(y)) \\
&= p(x|y)p(y|\mathcal{S}(y)) + \sum_i p(x|y_i^{ur})p(y_i^{ur}|\mathcal{S}(y)) + \sum_i p(x|y_i^o)p(y_i^o|\mathcal{S}(y)) \\
&\propto p(x|y)p(y|\mathcal{S}(y)) + \epsilon_1 + \epsilon_2
\end{aligned}
\tag{7}
$$

On the basis of Eq. 7 and Lemma 4, for any reply $y$ not belonging to universal replies, the Eq. 6 can be further deducted as:

$$log\,p(y|\mathcal{S}(y), x) \propto log\,p(y|\mathcal{S}(y)) + log\frac{p(x|y)}{p(x|y)p(y|\mathcal{S}(y)) + \epsilon} \propto log\frac{p(y|\mathcal{S}(y))}{p(y|\mathcal{S}(y)) + \mathcal{K}\epsilon} \quad (8)$$

where $\epsilon = \epsilon_1 + \epsilon_2 > 0$, which is also a sufficiently small positive value. Thus, optimizing the *word ordering probability* for the non-universal replies is partially equivalent to maximizing $p(y|\mathcal{S}(y))$. In fact the term $p(y|\mathcal{S}(y))$ is the language model probability and it is irrelevant with the query $x$ (Maning et al., 2009). In the sequential models, it is performed as $\prod_t p(y_t|y_{1:t-1}, \mathcal{S}(y))$, in other words the sequences are generated based only on previously outputted words. This equation indicates that optimizing the mainly seeks the grammatical competence based on the selected words.

## 2.4 BRIEF SUMMARY

In conclusion, the insufficient constraint of the target words' cross-entropy loss in NRG is the primary reason that hinders seq2seq models from exploring presumable parameters. This situation is mainly caused by the particular distribution of NRG corpus, since there exist many universal replies composed of high-frequent words in corpus. Consequently, the model tends to promotes such universal replies, regardless of the given query.

## 3 MAX-MARGINAL RANKING REGULARIZATION

As discussed above, various responses corresponding to the same query appearing in the training data leads to the undesired preference of NRG on universal replies, so an intuitive solution is removing the multiple replies and just keeping one-to-one pairs. However, filtering the training dataset in large scale raises the difficulty of model training. Besides, naively removing the multiple replies is detrimental to the reply diversity, which is important in NRG task. As shown in Table 1, an ideal chatbot agent is prospected to provide all listed replies and build a connection with some keywords such as 'film', 'background', 'director' and 'book', rather than other commonly appeared words like 'I', 'him', 'a' and 'really'.

Thus, under this assumption, we propose a max-marginal ranking loss to emphasize the queries' impact on these less common but relevant words. During training, as it becomes a necessity to constrain the learned feature space and reinforce related replies with more discriminative information, we classify the candidate responses into two categories: positive (i.e., highly related) and negative (i.e., irrelevant) answers. A training instance is re-constructed as a triplet $(x, y, y^-)$, where a tuple $(x, y)$ is the original query-response pair and noise $y^-$ is uniformly sampled from all of the responses in the training data. Given that, the model's loss function is reconstructed as:

$$\ell_\theta = -\log p(y|x) + \lambda \max\{0, -\log p(y|x) + \log p(y^-|x) + \gamma\} \quad (9)$$

where $\gamma > 0$, $\log p(y|x)$ denotes the cross-entropy loss between the model's prediction and ground truth sequences, and the second part encourages the separation between the irrelevant responses and related replies. Moreover, the hyper-parameter $\lambda$ defines the penalty for the seq2seq loss, it offers a degree of freedom to control the importance of the max-marginal between the positive and negative instances. The model is trained in the same setting as the conventional model when $\lambda = 0$.

The gradient of $\ell_\theta$ is computed using the sub-gradient method, as the second term is non-differentiable but convex (Agarwal & Collins, 2010). Supposing $\log p(y|x) - \log p(y^-|x) \leq \gamma$, the gradient of the composed loss function can be formalized as:

$$\nabla_\theta \ell_\theta = -\nabla_\theta \log p(y|x), \quad (10)$$

If $\log p(y|x) - \log p(y^-|x) > \gamma$, then the gradient should be written as:

$$\nabla_\theta \ell_\theta = -(\lambda + 1)\nabla_\theta \log p(y|x) + \lambda\nabla_\theta \log p(y^-|x). \quad (11)$$

The underlying motivation of our proposed loss function is based on three considerations: 1) Universal replies are more likely to be sampled from a statistical perspective, so adding a negative term would directly ease the weight of these generic responses, and the ranking regularization can penalize those irrelevant responses; 2) Positive and negative sentences overall share a same set of generic words, which suggests that the loss optimization should pay more attention on those different words rather than generic ones; 3) Only differentiable loss can solely be served as the model's

Table 2: Dataset statistics. For multiple replies, the three values represent the percentages of queries with one, two, and more than two responses, respectively. For the out of vocabulary (OOV) columns, the number in front of "/" denotes the percentage rate of the query, and the other one denotes replies.

|  | # train | # valid | # test |
| --- | --- | --- | --- |
| QA Pairs | 5,982,868 | 315,136 | 315,136 |
| Unique Replies | 4,499,176 | 298,723 | 287,312 |
| Multi Replies(%) | 70/24/6 | 97/2/1 | 96/3/1 |
| OOV (%) | .90/.90 | .92/.93 | .91/.92 |
| Vocab Size | | 29241/27859 | |

optimization goal for the sequence generation model. Furthermore, the newly proposed loss aims to penalize frequent words and irrelevant candidates, rather than repudiating the literal expression included in negative samples. Consequently, based on these considerations, we propose this term as a regularization to constrain the search space of parameters instead of the stand-alone loss function.

## 4 EXPERIMENTAL STUDIES

### 4.1 EXPERIMENTAL SETUPS

#### 4.1.1 DATASET DESCRIPTION

The dataset used in this study contained almost ten million query and response pairs collected from a popular Chinese social media site: Douban Group Chat[1]. All case studies used in this paper were extracted from this dataset and translated into English.

For easier training and better efficiency, the maximal lengths of queries and replies were set to 30 and 50 respectively. In all of our experiments, our dataset was split into the training, validation and test sets, with detailed statistical characterization given in Table 2. Thirty percent of queries had more than one responses, and each answer appeared about 1.33 times in the training dataset, which is consistent with our hypothesis in the analysis section.

#### 4.1.2 BASELINE MODELS

To validate the performance of the proposed model, the following baselines were considered:

- S2SA: The basic seq2seq model with attention mechanism (Bahdanau et al., 2015) at the target output side.

- S2SA + MMI: The best performing model in Li et al. (2016b) with the length norm based on the same S2SA.

- Ranking-Reg: The seq2seq model with proposed ranking regularization and attention. In this model, negative samples were uniformly sampled from the corpus, and the process was repeated 4 times for every positive case. The averaged negative loss was calculated as the probability of universal replies.

- Ranking-Reg + MMI: Ranking-Reg with MMI during inference procedure.

#### 4.1.3 EVALUATION METRICS

The quality of response was measured using both numeric metrics and human annotators. Firstly, *Word Perplexity* (PPL) is used to measure the model's ability to account for the syntactic structure for each utterance (Serban et al., 2016). Secondly, ROGUE score (Lin, 2004), which evaluates the extent of overlapping words between the ground-truth and predicted replies, was also adopted in experiments. Thirdly, we employed the widely used diversity measurements Distinct-1 and Distinct-2 to evaluate the number of distinct Unigrams and Bigrams of generated responses (Li et al., 2016b).

Furthermore, we recruited three highly educated human annotators to cross verify the quality of generated responses. We randomly sampled 100 queries and generated 10 replies for each query

---

[1]https://www.douban.com/group/explore

Table 3: Summarized results of testing set with metrics: Human Label, ROGUE-1, ROGUE-L, Distinct-1, Distinct-2 and PPL.

| Methods | Human Label | | | ROUGE | | Distinct | | PPL |
|---|---|---|---|---|---|---|---|---|
| | 0 | 1 | 2 | ROUGE-1 | ROUGE-L | 1 | 2 | |
| S2SA | 52.46% | **20.52%** | 27.02% | **4.97%** | **3.13%** | .129 | .285 | 110.0 |
| S2SA + MMI | 51.88% | 19.92% | 28.20% | 3.96% | 2.77% | .140 | .312 | 110.0 |
| Rank-Reg | 48.20% | 15.38% | **36.42%** | 3.45% | 2.55% | .163 | **.358** | **85.6** |
| Rank-Reg + MMI | _47.40%_ | 18.75% | 33.85% | 3.43% | 2.63% | **.167** | .345 | **85.6** |

using different models, with beam size set to 10. The labeled results were categorized into three degree (Xing et al., 2017; Mou et al., 2016):

**0**: The response cannot be used as a reply to the message. It is either semantically irrelevant or not fluent (e.g., with grammatical errors or UNK).

**1**: The response can be used as a reply to the message, which includes the universal replies such as "Yes, I see" , "Me too" and "I dont know".

**2**: The response is not only relevant and natural, but also informative and interesting.

### 4.1.4 TRAINING PROCEDURES

For all of the models, LSTM was chosen as the recurrent cell, and there were 512 hidden units for both the encoder and decoder (Greff et al., 2017). Embedding size and batch size were set to 200 and 20 respectively. The Adam algorithm was employed for gradient optimization (Kingma & Ba, 2015), and the initial learning rate was 1e-4. All of the models were implemented in Theano (Theano Development Team, 2016), and each ran on a standalone K40m GPU device for 7 epochs, which took 7 days; twice longer time was required for training models with rank regularization.

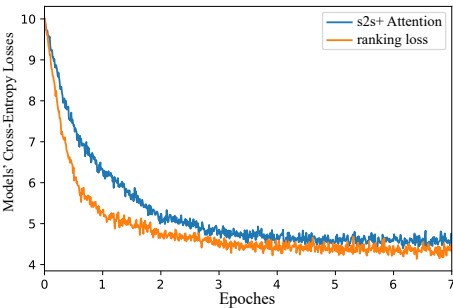

Figure 2: Learning curve for the two models.

The last two models with the rank regularization share the related hyper-parameters. We set $\lambda$ to 0.1 and $\gamma$ to 0.18, according to the model's performance on the validation set.

Fig. 2 shows cross-entropy loss flows vs. training epoch numbers. The model with max-marginal ranking regularization converges faster than S2SA throughout the training. This shows that the additional regularization term helps to speed up the fitting by removing these sub-optimal paths.

## 4.2 RESULTS AND ANALYSIS

### 4.2.1 EXPERIMENTAL RESULTS.

The performance of four models on existing metrics is summarized in Table 3. The model with the max-marginal ranking regularization outperforms the model with primary loss function on the target loss PPL. As the MMI method is performing during inference, losses of models with MMI are identical to those without revision.

However, the results are opposite regarding the ROGUE scores. The generated responses by the S2SA model contain more words appearing in the ground truth answers. These experimental results can be attributed to mainly two factors. a) The very low ROUGE scores reflect few words shared by any predictions and the ground truth. Most n-gram overlaps belonging to the common words, such as "I", "are", "that". b) A certain proportion of replies in the test set are universal themselves. Therefore, S2SA has achieved higher ROUGE score as its' results are more consistent with those common ground truth responses.

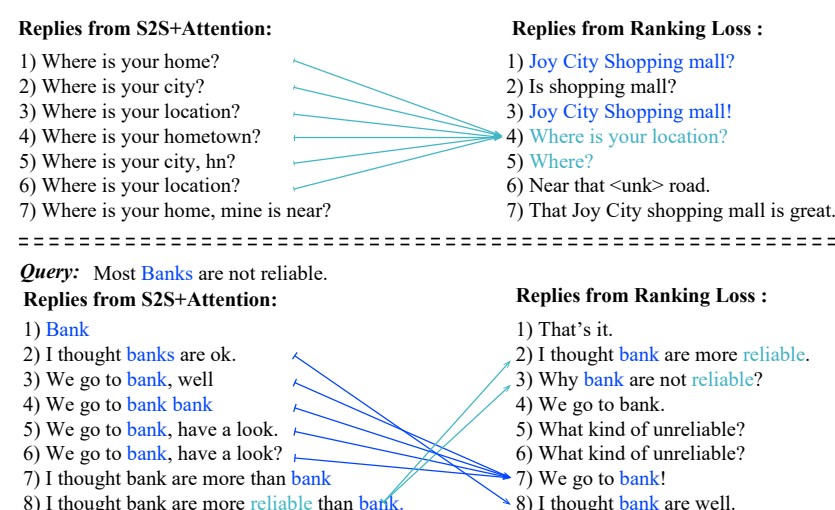

Figure 3: Response re-rank capability. Responses generated by the basic model and model with rank loss are linked by arrows, and same topics are typeset using the same color. Some ungrammatical and incomprehensible sentences exist due to the translating try to keep the word order.

The human evaluation is the most important metric, and it is clear from Table 3 that the models with rank regularization beat S2SA with a large margin. It increases the number of meaningful responses by around 10% and reduces the number of irrelevant cases by around 4%. Meanwhile, most the acceptable replies (labeled as "1" or "2") of S2SA is labeled as "1", which indicates the model prefer the safe responses. We attribute the gaps to the promotion of highly related words and reducing of the universal replies. Same trend can be also spotted on Distinct-1 and Distinct-2, it reveals the model's ability to generate diverse responses (Li et al., 2016b; Serban et al., 2015). The seq2seq model yields lower levels of unigram and bigram diversity than the rank loss model.

As another comparison, we note that the improvement introduced by MMI is much smaller than that introduced by the ranking regularization, whereas MMI is a widely used mechanism for promoting diverse responses during inference. Besides, performing it upon the regularization reduces the rate of informative and interesting responses. This observation indicates that the fundamental reason behind generating tasteless or inappropriate replies is that Seq2Seq model learned from conversational corpora prefers universal replies. Moreover, the revision during the greedy search is less effective on solving the underlying problems than the proposed ranking regularization.

### 4.2.2 RANKING LOSS FOR GENERIC RESPONSES.

From the generated results, it is found that the seq2seq model with the ranking regularization term prefers meaningful content when the query contains sufficient amount of information. We present top responses for two queries generated by different models in Fig. 3. As shown in the first case, user posts a query which initiates a complicated discussion about locations. It is observed that S2SA converges to a typical "where is your" pattern of replies when discussing locations, which is an example of universal replies. As the greedy beam search strategy is utilized during inference, many location-related constraints further promote these relevant universal replies instead of more varied results from different beams. In contrast, some of the responses in the right column captured the "commercial street" clues and inferred a possible location "Joy City shopping mall" demoting the generic beams results. We attributed this to the boosting ability associated with semantically relevant words, as mentioned in Section 3.

The second case is quite different. In this case, the seq2seq model did not perform satisfactorily. Even though the subject "bank" was extracted into the generated candidates, we cannot perceive the results aligned with the same "not reliable" topic, and most of them were just chosen from two beams. Inspecting the replies generated by the rank loss model, we found that more complicated and

diverse sentences that discuss "unreliable" can be generated, and irrelevant answers about "bank" are lower-ranked. To further investigate the difference brought by the max-marginal ranking regularization, we randomly sampled more cases shown in the Fig. 4 as appendix. Even though some of them were bad cases and contained some grammatical errors, overall the model with rank regularization tends to generate more informative and interesting sentences compared with baselines.

In conclusion, the seq2seq model with rank regularization can not only formulate the conditional language model but also boost related answers to higher ranks than the rest of universal or inappropriate replies.

## 5 RELATED WORK

Recent years have witnessed the rapid development of data-driven dialog models with the help of accumulated conversational data from online communities. Query-response pairs are modeled by Seq2Seq models with attention mechanism (Sutskever et al., 2014; Serban et al., 2016; Bahdanau et al., 2015), and NRG model are designed to maximize the likelihood of target response given the source query. As there exist various reasonable responses given a query, some researches conclude that the limited information in many queries constrains the model inference, which makes the NRG models prefer universal replies (Shao et al., 2017; Mou et al., 2016).

To address this issue, various works are conducted on bringing more information to Seq2Seq models. Some works focus on constraining the replies with topic information or keywords (Mou et al., 2016; Xing et al., 2017; Wang et al., 2017; Wu et al., 2018). Other researchers argue that diverse responses are buried by the greedy beam-search rules (Li et al., 2016b), so their works mainly focus on involving more punishment or randomness in the inference stages. For example, Li et al. (2016b) constrain the search space using mutual information with the query, while Shao et al. (2017) randomly chose candidate words from top beams to constrain short phrases. These existing works mainly focus on the generation strategies during inference, in contrast, the model's architecture and loss function have rarely been explored.

Serban et al. (2017) introduce to model the underlying distribution over possible replies directly with supposing various latent variables to affect the response generation. Shen et al. (2017) further constructs a variational lower bound for response constraint. During inference, these models generate responses by first sampling an assignment of latent variables, so that models can generate more diverse responses. Such methods attempt to improve the diversity of responses by modifying the Seq2Seq architecture, and our analysis may be also helpful to design more effective latent variable based models to restrain current problems. Besides, the ranking penalty has also been used by Wiseman & Rush (2016), they employ a word-level margin to promote ground-truth sequences appearing in the beam search results. Different from our method, they directly optimize the beam search procedure to fine-tune the trained model.

## 6 CONCLUSION

Eliminating generic responses is the essence for the widely practical utilization of the Seq2Seq based neural response generation architectures, and thus, this paper has conducted a thorough investigation on the cause of such uninformative responses and proposed the solution from the statistical perspective. The main contributions of this work can be summarized as follows: a) The theoretical analysis is performed to capture the root reason of NRG models producing generic responses through the optimization goal of models and the statistical characteristics of human-to-human conversational corpora, which has been little studied currently. In detail, we have decomposed the goal of NRG into the optimizations of word selection and word ordering, and finally derived that NRG models tend to select common words as responses and order words from the language model perspective which ignores queries. b) According to the analysis, a max-marginal ranking regularization term is proposed to cooperate with the learning target of Seq2Seq, so as to help NRG models converge to the status of producing informative responses, rather than merely manipulating the decoding procedure to constrain the generation of universal replies. Furthermore, the empirical experiments on the conversation dataset indicate that the models utilizing this strategy notably outperform the current baseline models.

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

## A CASES

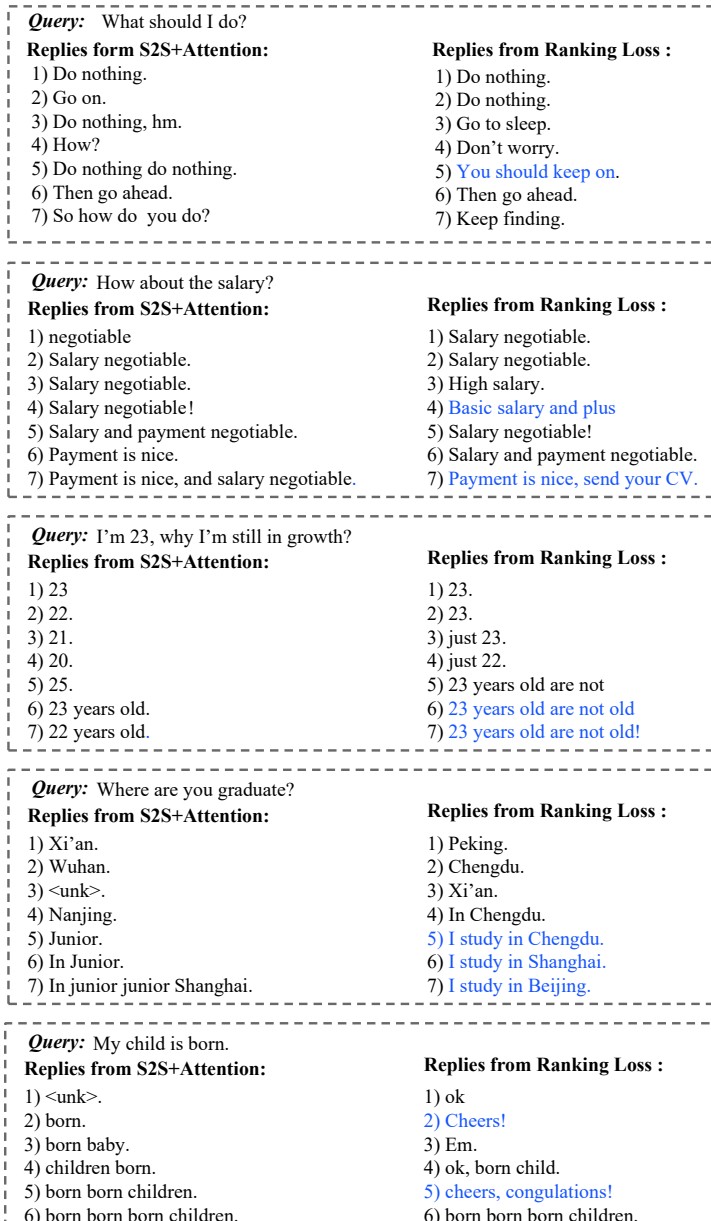

**Query:** What should I do?

| **Replies form S2S+Attention:** | **Replies from Ranking Loss :** |
|---|---|
| 1) Do nothing. | 1) Do nothing. |
| 2) Go on. | 2) Do nothing. |
| 3) Do nothing, hm. | 3) Go to sleep. |
| 4) How? | 4) Don't worry. |
| 5) Do nothing do nothing. | 5) You should keep on. |
| 6) Then go ahead. | 6) Then go ahead. |
| 7) So how do you do? | 7) Keep finding. |

**Query:** How about the salary?

| **Replies from S2S+Attention:** | **Replies from Ranking Loss :** |
|---|---|
| 1) negotiable | 1) Salary negotiable. |
| 2) Salary negotiable. | 2) Salary negotiable. |
| 3) Salary negotiable. | 3) High salary. |
| 4) Salary negotiable! | 4) Basic salary and plus |
| 5) Salary and payment negotiable. | 5) Salary negotiable! |
| 6) Payment is nice. | 6) Salary and payment negotiable. |
| 7) Payment is nice, and salary negotiable. | 7) Payment is nice, send your CV. |

**Query:** I'm 23, why I'm still in growth?

| **Replies from S2S+Attention:** | **Replies from Ranking Loss :** |
|---|---|
| 1) 23 | 1) 23. |
| 2) 22. | 2) 23. |
| 3) 21. | 3) just 23. |
| 4) 20. | 4) just 22. |
| 5) 25. | 5) 23 years old are not |
| 6) 23 years old. | 6) 23 years old are not old |
| 7) 22 years old. | 7) 23 years old are not old! |

**Query:** Where are you graduate?

| **Replies from S2S+Attention:** | **Replies from Ranking Loss :** |
|---|---|
| 1) Xi'an. | 1) Peking. |
| 2) Wuhan. | 2) Chengdu. |
| 3) <unk>. | 3) Xi'an. |
| 4) Nanjing. | 4) In Chengdu. |
| 5) Junior. | 5) I study in Chengdu. |
| 6) In Junior. | 6) I study in Shanghai. |
| 7) In junior junior Shanghai. | 7) I study in Beijing. |

**Query:** My child is born.

| **Replies from S2S+Attention:** | **Replies from Ranking Loss :** |
|---|---|
| 1) <unk>. | 1) ok |
| 2) born. | 2) Cheers! |
| 3) born baby. | 3) Em. |
| 4) children born. | 4) ok, born child. |
| 5) born born children. | 5) cheers, congulations! |
| 6) born born born children. | 6) born born born children. |
| 7) born children born children. | 7) born children born children. |

Figure 4: Cases for comparing the S2SA and the model with ranking regularization, and the topics or expressions of the generated replies marked with blue are excluded in the responses generated by SASA.

