# OpenReview forum: "Why Do Neural Response Generation Models Prefer Universal Replies?"
_ICLR.cc/2019/Conference_

### Official Review · AnonReviewer2 · 2018-10-26
**Technically unsound work**

**Rating:** 1
**Confidence:** 5

**Review:**

This paper presents a framework for understanding why seq2seq neural response generators prefer "universal"/generic replies. To do so the paper breaks down the response generation probability into the probability of selecting the set of tokens (reflecting the topic of the output) and then selecting an ordering of the tokens (reflecting the syntax of the output.)

The results presented in this paper are not technically sound. E.g the derivation in Eq(2) derive a meaningless bound. Here is why:
1. The first equality assumes that the words in a set are independent which is not true.
2. In the second equality, the authors incorrectly replace the summation of word probability in each sentence with the summation of word probabilities over all unique words (the set) overall sentences. This is simply not true if there are common words shared between sentences.
3. Perhaps the biggest issue is the incorrect application of Jensen's lemma. JL is often used as
log(\sum_i a_i x_i) > \sum_i a_i log x_i if \sum_i a_i = 1. Instead what authors have used is
log (\sum_i x_i) > \sum_i log(x_i), which is not always true, and is trivially true for all x_i < 1. In fact, this bound is not even tight (unlike Jensen's lemma) and the *worst* part is that the LHS increases if we add more x_i (<1) and the RHS decreases. This means this bound is far from being meaningful and as such should be summarily ignored.

Similarly, in section 2.3, the technical content is quite poor. Why is this true -- "the amount of possible queries M of y... 1 << M \propto N"? There are many assumptions in lemma 3 that are quite difficult to unpack to verify the correctness e.g. can the most frequent words not occur at all in "non-universal" replies? I am not going more into the details in this section because I think the problems with section 2.2 are themselves dealbreakers.

Overall, given the problems this work is not technically sound to be accepted.

---

> ### Author Response · Authors · 2018-11-13
> **Response to the comments of Reviewer2**
>
> Thank you for your helpful comments.
>
> For some questions mentioned in the review comment, we would like to make the clarification as follows:
>
> Q1: “perhaps the biggest issue is the incorrect application of Jensen’s lemma”
> As detailed in another comment given by us, Equation 2 is actually indirectly derived using Jensen’s Inequality though it still holds.
>
> For more details, the size of \cup_k^K{S(y_k)}, noted as L_S, should be included in Equation 2 when performing Jensen’s Inequality, where L_S <= K*T, and K is the number of responses of x and T is the fixed number of words in a response. Then the last inequality in Equation 2 can be derived as follows.
> \sum \log p(w|x) <= L_S \log \sum[p(w|x) / L_S]
> To simplify, we use \sum to denote \sum_{}^{}.
>
> Then, the Eq. 3 analyzes the last term in the previous inequality:
> L_{S} \log \sum_{w \in \cup_k^K S(y_k)} \frac{p(w|x)}{L_{S}}
> = L_{S} \log \frac{E(w|x) * T}{K*T*L_{S}}
> \propto \log \frac{1}{(K*L_{S})^{L_{S}}}
> \leq \frac{1}{K^{2K}}
> which is much smaller than 1 / K, so that the analysis still holds.
>
> We have already updated this part to the right version in the revised manuscript.
>
> Q2: “The first equality assumes that the words in a set are independent which is not true.”
> Actually, the independent hypothesis is often used in the natural language generation related task (Hashimoto et al. 2018) especially in the statistical machine translation (Qch, et al. 2002, Koehn et al., 2003), neural machine translation (Weng et al. 2017), as well as previous NRG related works (Wu et al. 2018). Such a hypothesis can be taken to simplify and analyze the complicated natural language processing problem, which is similar to some other hypotheses such as the Markov Chain.
> Besides, the model aims to maximize the generation probability of each {y_k} in the training procedure, but only produces a single one in the inference phase. Considering this fact, most of the words (appearing in different sentences) could be taken as independent naturally.
>
> Q3: “…e.g. can the most frequent words not occur at in “non-universal” replies?”
> As claimed in the paper, the very first characteristic of a universal reply can be described as “A response is universal if it consists of only top-t ranked words”, which is not equivalent to “top-t ranked words only exist in the universal replies”. It should be noted that Equation 4 is derived based on this characteristic. Moreover, we totally agree with the fact that the “the most frequent words not occur at in “non-universal” replies” is not true.
>
> Q4: ”Why is this true – “The amount of possible queries M of y is directly proportional to the size of query-response pairs N, noted as 1 << M \propto N.””
> NRG models tend to generate universal replies, such as “I don’t know”, “I’m OK” etc., and this trend is one of the most tough problem faced by current NRG models (Sordoni et al., 2015; Vinyals and Le, 2015; Li et al., 2016b;c;d; Li & Jurafsky, 2016; Mou et al., 2016; Xing et al., 2017; Shao et al., 2016; 2017; Serban et al., 2017). Moreover, there are some other aliases of universal replies including “safe responses”, “common replies”, and “generic responses”, which indicates these responses are mostly high-frequent in the corpus.
> Especially, as discussed in the previous studies, the universal reply can be the subsequent utterance of a rather large number of varieties of queries (Li et al., 2016a, Li et al., 2016b). Thus, given the large-scale corpus of query-reply pairs, the number of universal replies M is apparently much larger than 1. Meanwhile, with the unchanged data distribution, if we sample a k-times larger corpus from the source, the number of universal replies will be extended to k-times larger, that is the reason we claim “M \propto N”. Furthermore, “M \propto N” is not equivalent to claiming that M is as large as N.

---

> > ### Author Response · Authors · 2018-11-13
> > **References**
> >
> > Kazuma Hashimoto and Yoshimasa Tsuruoka. Accelerated reinforcement learning for sentence generation by vocabulary prediction. arXiv preprint arXiv:1809.01694, 2018.
> >
> > Philipp Koehn, Franz Josef Och, and Daniel Marcu. Statistical phrase-based translation. In Proceedings of the 2003 Conference of the North American Chapter of the Association for Computational Linguistics on Human Language Technology Volume 1, pp. 48–54. Association for Computational Linguistics, 2003.
> >
> > Chaozhuo Li, Yu Wu, Wei Wu, Chen Xing, Zhoujun Li, and Ming Zhou. Detecting context dependent messages in a conversational environment. In Proc. of COLING, pp. 1990–1999, 2016a.
> >
> > Jiwei Li, Michel Galley, Chris Brockett, Jianfeng Gao, and Bill Dolan. A diversity-promoting objective function for neural conversation models. In Proc. of NAACL-HLT, pp. 110–119, 2016b.
> >
> > Jiwei Li, Michel Galley, Chris Brockett, Georgios P. Spithourakis, Jianfeng Gao, and William B.Dolan. A persona-based neural conversation model. InProc. of ACL, pp. 994–1003, 2016c.
> >
> > Jiwei Li, Will Monroe, and Dan Jurafsky. A simple, fast diverse decoding algorithm for neural generation. CoRR, abs/1611.08562, 2016d.
> >
> > Lili Mou, Yiping Song, Rui Yan, Ge Li, Lu Zhang, and Zhi Jin. Sequence to backward and forward sequences: A content-introducing approach to generative short-text conversation. In Proc. of COLING, pp. 3349–3358, 2016.
> >
> > Franz Josef Och and Hermann Ney. Discriminative training and maximum entropy models for statistical machine translation. In Proceedings of the 40th annual meeting on association for computational linguistics, pp. 295–302. Association for Computational Linguistics, 2002.
> >
> > Iulian Vlad Serban, Alessandro Sordoni, Yoshua Bengio, Aaron C. Courville, and Joelle Pineau. Building end-to-end dialogue systems using generative hierarchical neural network models. In Proc. of AAAI, pp. 3776–3784, 2016.
> >
> > Iulian Vlad Serban, Alessandro Sordoni, Ryan Lowe, Laurent Charlin, Joelle Pineau, Aaron C Courville, and Yoshua Bengio. A hierarchical latent variable encoder-decoder model for generating dialogues. In AAAI, pp. 3295–3301, 2017.
> >
> > Yuanlong Shao, Stephan Gouws, Denny Britz, Anna Goldie, Brian Strope, and Ray Kurzweil. Generating high-quality and informative conversation responses with sequence-to-sequence models. In Proc. of EMNLP, pp. 2210–2219, 2017.
> >
> > Alessandro Sordoni, Michel Galley, Michael Auli, Chris Brockett, Yangfeng Ji, Margaret Mitchell, Jian-Yun Nie, Jianfeng Gao, and Bill Dolan. A neural network approach to context-sensitive generation of conversational responses. arXiv preprint arXiv:1506.06714, 2015.
> >
> > Oriol Vinyals and Quoc V. Le. A neural conversational model. arXiv preprint arXiv:1506.05869, 2015.
> >
> > Rongxiang Weng, Shujian Huang, Zaixiang Zheng, XIN-YU DAI, and CHEN Jiajun. Neural machine translation with word predictions. In Proceedings of the 2017 Conference on Empirical Methods in Natural Language Processing, pp. 136–145, 2017.
> >
> > Yu Wu, Wei Wu, Zhoujun Li, Can Xu, and Dejian Yang. Neural response generation with dynamic vocabularies. national conference on artificial intelligence, 2018.
> >
> > Chen Xing, Wei Wu, Yu Wu, Jie Liu, Yalou Huang, Ming Zhou, and Wei-Ying Ma. Topic aware neural response generation. In Proc. of AAAI, pp. 3351–3357, 2017.

---

> > ### Comment · AnonReviewer2 · 2018-11-26
> > **Unconvinced; the bounds presented are too loose to be useful**
> >
> > Thanks for responding to my review and updating your paper accordingly. However, the bounds in the paper still do not look useful.
> > e.g. in (3) you have
> > log (K × L_S )^{L_S} \leq log((1/K)^{1/2K}).
> > I am assuming this follows if we assume  K \leq L_S; why is that necessarily true? Theoretically it's possible to have the opposite while in practice K << L_S. In the latter case, the bound is not very tight to be useful.
> >
> > Similarly your reason for “M \propto N” is tautological -- you're creating the corpus by sampling a constant number of times from source sentences and claim that the resulting output size is proportional to N -- this falls out of your sampling assumption. This may not be true for all distribution assumptions.

---

> > > ### Author Response · Authors · 2018-11-27
> > > **Response to the comment “Unconvinced; the bounds presented are too loose to be useful”**
> > >
> > > Thank you for the insightful comments. Actually, K \leq L_S is not rigorous theoretical, as there may exist a few cases not satisfying this inequation.
> > > Nevertheless, we can claim that, given the obvious condition “1 \leq L_S”, the upper bound in Equation 3 is \log frac{1}{K}, and the analysis still holds. We have fixed this issue in the revised version.
> > >
> > > For the second question about “M \propto N”, it should be noted that our corpus is composed of crawled real human conversations without any tailor-made sampling strategies, and the sampling operation is only conducted to get the negative responses, which does not influence the distribution of the corpus indeed.

---

### Official Review · AnonReviewer1 · 2018-11-03
**main contribution: improving neural response by de-emphasizing universal responses by modifying loss/training**

**Rating:** 7
**Confidence:** 3

**Review:**

The paper looks into improving the neural response generation task by deemphasizing the common responses using modification of the loss function and presentation the common/universal responses during the training phase. The authors show that the approach yields better results in the dataset considered using various measures and human evaluation.

Improvement Points
- the explanation for low ROGUE measure due to the method favoring non-repetitive words sounds like it can be supported using numerical statistics, than hand-waiving argument
- for the timing, how much time was taken to tune the additional parameters (how the # of negative responses sampled for each positive response was chosen as four via uniform sampling)
- some description about
a) how many users are there, what type of conversation/active users/topics etc.
b) what time frame was used during data collection (this may have implications for lemma asserting zipf)
- it would be interesting to know for the trivial questions if the performance was impacted by the deemphasizing (one that do result in universal replies)

---

> ### Author Response · Authors · 2018-11-13
> **Response to the comments of Reviewer1**
>
> Thank you for your helpful comments and suggestions. We will add more details about the data and statistics in the appendix according to your suggestions.
>
> For some questions mentioned in the review comment, we would like to make the clarification as follows:
>
> Q1: “for the timing, how much time was taken to tune the additional parameters (how the # of negative responses sampled for each positive response was chosen as four via uniform sampling)”
> As mentioned in the question, the encoding-decoding process of our method is around 8 times longer than the standard S2SA for a ground-truth query-reply pair. However, since the encoder modeling the query x in learning each (x, y, y^{-}) pair can be shared, the final training cost is around 6 times of the standard S2SA.
> Moreover, the proposed max-margin regularization can be used to fine-tune a trained S2SA model, and our experiment shows the model can converge in only 1-2 fine-tuning epochs. Therefore, in practice our proposed method can be more efficient.
>
> Q2: “some description about
> a) how many users are there, what type of conversation/active users/topics etc.
> b) what time frame was used during data collection (this may have implications for lemma asserting zipf)”
> The dataset is extracted from 1 million conversations within half year. The source of our dataset is Douban group, which is a forum similar to Reddit. We did not restrict the types of communities, thus the collected conversations cover various topics including games, movie, music, sports, etc. The approximated number of users is around 1 million according to the statistics on small samples.
>
> Q3: “it would be interesting to know for the trivial questions if the performance was impacted by the deemphasizing (one that do result in universal replies)”
> It is really an interesting question. For each query, the sampled negative replies are not equivalent to the ground-truth, and we do not expect the model to learn the ground-truth pairs with universal replies. The proposed loss function aims to punish those universal replies from a statistical perspective, by considering the distributions of words and sentences. Therefore, the regularization does not directly influence those trivial questions.

---

### Official Review · AnonReviewer3 · 2018-11-04
**Interesting problem but extremely bad execution**

**Rating:** 3
**Confidence:** 4

**Review:**

The paper investigates the problem of universal replies plaguing the Seq2Seq neural generation models. The problem is indeed quite important because for problems with high entropy solutions the seq2seq models have been shown to struggle in past literature. While the authors do pick a good problem, that's where the quality of the paper ends for me. The paper goes on an endless meandering through a lot of meaningless probabilistic arguments.  First of all, factorizing a seq2seq model as done in equation 1 is plain wrong. The model doesn't operate by first selecting a set of words and then ordering them. On top of this wrong factorization, section 2.2 & 2.3 derives a bunch of meaningless lemmas with extremely crude assumptions. For example, for lemma 3, M is supposed to be some universal constant defined to be the frequency of universal replies while all other replies seem to have a frequency of 1. Somehow through this wrong factorization and some probabilistic jugglery, we arrive at section 3 where the takeaway from section 2 is the rather known one that the model promotes universal replies regardless of query.

In section 3, the authors then introduce the "max-marginal regularization" which is a linear combination of log-likelihood and max-margin (where the score is given by log-likelihood) losses. Firstly, the use of word "marginal" instead of "margin" seems quite wrong to say the least.  Secondly, the stated definition seems to be wrong. In the definition the range of values for \gamma is not stated. I consider the two mutual exclusive and exhaustive cases (assuming \gamma not equals 0) below and show that both have issues:
(a) \gamma > 0: This seems to imply that when the log-likelihood of ground-truth is already \gamma better than the log-likelihood of the random negative, the loss comes to life. Strange!
(b) \gamma < 0: This is again weird and doesn't seem to be the intended behavior from a max-margin{al} loss.
I'm assuming the authors swapped y with y^{-} in the "regularization" part.
Anyways, the loss/regularization doesn't seem to be novel and should have been compared against pure max-margin methods as well.

Coming to the results section, figure 3 doesn't inspire much confidence in the results. For the first example in figure 3, the baseline outputs seem much better than the proposed model, even if they follow a trend, it's much better than the ungrammatical and incomprehensible sentences generated by the proposed model. Also there seems to be a discrepancy in figure 3 with the baseline output for first query having two "Where is your location?" outputs.  The human column of results for Table 3 is calculated over just 100 examples which seems quite low for any meaningful statistical comparison. Moreover, not quite sure why the results used the top-10 results of beam instead of the top-1.

A lot of typos/wrong phrasing/wrong claims and here are some of them:
(a) Page 1, "lead to the misrecognition of those common replies as grammatically corrected patterns"? - No idea what the authors meant.
(b) Page 1, "unconsciously preferred" - I would avoid attaching consciousness before AGI strikes us.
(c) Page 1, "Above characters" -> "Above characteristics"
(d) Page 1, "most historical" -> "most previous"
(e) Page 2, "rest replies" -> "rest of the replies"
(f) Page 3, "variational upper bound" -> Not sure what's variational about the bound
(g) "Word Perplexity (PPL) was used to determine the semantic context of phrase-level utterance"? - No idea what the authors meant.

---

> ### Author Response · Authors · 2018-11-13
> **Response to the comments of Reviewer3**
>
> Thank you for your helpful comments. We will improve our paper accordingly.
>
> We would like to clarify some points mentioned in the review comment, as follows:
>
> Q1: “factorizing a seq2seq model as done in equation 1 is plain wrong”
> First, Equation 1 performs the decomposition on the loss function defined for NRG task, which is in fact widely used in the natural language generation related tasks (Hashimoto et al. 2018), especially in the statistical machine translation (Qch, et al. 2002, Koehn et al., 2003), neural machine translation (Weng et al. 2017), as well as previous NRG related works (Wu et al. 2018).
> It should be noted that, the decomposition in Equation 1 is indeed not about the training procedure of the seq2seq models for NRG. Moreover, from the perspective of probability theory, any event z (either latent or explicit) could be involved to replace S(y) as -\log p(y|x)= -\log p(z|x) -\log p(y|z,x) in Equation 1. In this paper, we assign the set of words of response as the event z to perform the analysis. The result of the analysis on -\log p(S(y)|x) -\log p(y|S(y),x) also holds for -\log p(y|x) and its decomposition with any other forms of z. Besides, the usages of this notation S(y) are consistent throughout Section 2.
>
> Q2: “…while all other replies seem to have a frequency of 1…”
> Actually, the frequencies of non-universal replies should be a positive constant C (1 <= C << M). To simplify, as mentioned in Section 2.3.1, we suppose that C equals to 1, but it is actually not rigorous. We have fixed it in the revised version.
> Consequently, Equation 5 based on this declare will be deducted as follows:
> \sum_i p(y^{ur}_i | S(y))
> = \frac{\sum_i^m f(y^{ur}_i)}{\sum_i^m f(y^{ur}_i) + \sum_i^{n-m} f(Y^{o}_i)}
> = \frac{M * m}{M * m + c * (n - m)}
> = \frac{M}{M + n / m - c}
> > \frac{M}{M + 3 - c}
> Thus, the conclusion of Lemma 3 still holds.
>
> Q3: “…the range of values for \gamma is not stated…”
> The \gamma should be larger than 0 and consistent with the standard max-margin loss (Agarwal & Collins, 2010; Hu et al, 2014). We have added its range in the revised manuscript. Besides, the discussion in the review comment about the case of \gamma > 0 can be established, since the preference of choosing more informative words is exactly what we expect the model to capture. Besides, since in the decoding phase, {y_1, … y_{i-1}} is given when predicting y_i, the log-likelihood of the ground-truth is not always higher than that of the random negative, in other words, the regularization does not often come to life.
> However, we believe that comparing our loss against the pure max-margin methods is not reasonable, since the max-margin loss also penalizes the literal expression (language modeling) included in negative samples, which makes the model fail to generate grammatical sentences. Based on this consideration, we propose this term as a regularization, which is also mentioned in the last paragraph of Section 3.
>
> Q4: “…ungrammatical and incomprehensible sentences generated by the proposed model … duplicated sentences in the output of the baseline model…”
> Sorry for the confusion caused by the representation in the original manuscript First, the samples in Figure 3 are the translated version from Chinese. The translation guideline is to keep as many patterns, key words within the Chinese sentence as possible. Therefore, some ungrammatical and incomprehensible sentences appear. Besides, the duplicated sentences are attributed to the translation as well, because there are no duplications in the untranslated group of generated sentences. We will fix this issue and provide the original Chinese sentences.
>
> Q5: “…100 examples which seems quite low for any meaningful statistical comparison ...”
> In fact, we randomly sample 100 query and generate 10 replies for each query, which produces 1000 examples for each model and 4000 examples for each labeler. It should be noted that, the human evaluation is a widely used measure in NRG to cooperate with the numeric metrics, and the count of manually labeled examples in previous studies always ranges from hundreds to thousands (Li et al., 2016a; Mou et al., 2016; Xu et al. 2017; Serban et al. 2017).
>
> Q6: “Moreover, not quite sure why the results used the top-10 results of beam instead of the top-1”
> A chatbot agent which can reply with diverse answers means such agent generates various responses for a fixed query, and these various responses (generated by beam search) do not share information or pattern between each other. However, as current NRG models tend to generate a series of universal replies from the same beam, they often fail to reply diverse answers.
> The objective of our paper is to investigate the reason for NRG models preferring universal replies and to propose a method to promote the diversity of generated responses from NRG models. Correspondingly, it is only reasonable to measure the performance on diversity by evaluating more than one responses for each query.

---

> > ### Author Response · Authors · 2018-11-13
> > **References**
> >
> > Shivani Agarwal and Michael Collins. Maximum margin ranking algorithms for information retrieval. In Proc. of ECIR, pp. 332–343, 2010.
> >
> > Kazuma Hashimoto and Yoshimasa Tsuruoka. Accelerated reinforcement learning for sentence generation by vocabulary prediction. arXiv preprint arXiv:1809.01694, 2018.
> >
> > Baotian Hu, Zhengdong Lu, Hang Li, and Qingcai Chen. Convolutional neural network architectures for matching natural language sentences. In Advances in neural information processing systems, pp. 2042–2050, 2014.
> >
> > Philipp Koehn, Franz Josef Och, and Daniel Marcu. Statistical phrase-based translation. In Proceedings of the 2003 Conference of the North American Chapter of the Association for Computational Linguistics on Human Language Technology Volume 1, pp. 48–54. Association for Computational Linguistics, 2003.
> >
> > Jiwei Li, Michel Galley, Chris Brockett, Jianfeng Gao, and Bill Dolan. A diversity-promoting objective function for neural conversation models. In Proc. of NAACL-HLT, pp. 110–119, 2016a.
> >
> > Lili Mou, Yiping Song, Rui Yan, Ge Li, Lu Zhang, and Zhi Jin. Sequence to backward and forward sequences: A content-introducing approach to generative short-text conversation. In Proc. of COLING, pp. 3349–3358, 2016.
> >
> > Franz Josef Och and Hermann Ney. Discriminative training and maximum entropy models for statistical machine translation. In Proceedings of the 40th annual meeting on association for computational linguistics, pp. 295–302. Association for Computational Linguistics, 2002.
> >
> > Iulian Vlad Serban, Alessandro Sordoni, Ryan Lowe, Laurent Charlin, Joelle Pineau, Aaron C Courville, and Yoshua Bengio. A hierarchical latent variable encoder-decoder model for generating dialogues. In AAAI, pp. 3295–3301, 2017.
> >
> > Yuanlong Shao, Stephan Gouws, Denny Britz, Anna Goldie, Brian Strope, and Ray Kurzweil. Generating high-quality and informative conversation responses with sequence-to-sequence models. In Proc. of EMNLP, pp. 2210–2219, 2017.
> >
> > Rongxiang Weng, Shujian Huang, Zaixiang Zheng, XIN-YU DAI, and CHEN Jiajun. Neural machine translation with word predictions. In Proceedings of the 2017 Conference on Empirical Methods in Natural Language Processing, pp. 136–145, 2017.
> >
> > Yu Wu, Wei Wu, Zhoujun Li, Can Xu, and Dejian Yang. Neural response generation with dynamic vocabularies. national conference on artificial intelligence, 2018.
> >
> > Zhen Xu, Bingquan Liu, Baoxun Wang, SUN Chengjie, Xiaolong Wang, Zhuoran Wang, and Chao Qi. Neural response generation via gan with an approximate embedding layer. In Proceedings of the 2017 Conference on Empirical Methods in Natural Language Processing, pp. 617–626, 2017.

---

> > ### Comment · AnonReviewer3 · 2018-11-19
> > **Still not convinced**
> >
> > Thanks for the detailed reply, I appreciate the effort.
> >
> > I agree with the fact that you can decompose the seq2seq loss as done in equation 1. The explanation you gave is indeed correct, except marginalization over the latent variable z which in case of set of words is unique and hence the equation indeed does hold. My concern however had more to do with the fact that such a decomposition didn't seem an ideal way of analyzing seq2seq models. Section 2.2 makes a very strong assumption that words are independent given x. This assumption just ruins all the analysis for me. I do see that in reply to other reviewers you mentioned that (Hashimoto, 2018) makes the same assumption. Well I don't see that from my reading of that paper. And even if some paper does make these assumptions, I'm not convinced. This fundamental disagreement is why I'm not going to update my rating.
> >
> > Thanks for updating Lemma 3.
> >
> > Thanks for updating the domain of gamma. However, the loss is still weird to me as I stated in my original review. Just analyzing the regularization term, the term is non-zero when the log-likelihood of ground truth is already at least gamma better than some negative output. This is quite strange to say the least. The positive values of gamma and lambda mean that the regularization seems to do the opposite of what is intended as described in the manuscript as "the second part encourages the separation between the irrelevant responses and related replies". Analyzing equation 11 with \lambda=0.1 (as used in experiments), means that in the updated loss:
> > (a) log-likelihood of ground-truth is updated with a reduced weight of 0.9 from 1.0
> > (b) log-likelihood of negative sample is encouraged with a weight of 0.1
> > I feel the max term requires sign swapping but I'll leave it to the authors to figure this out.
> >
> > Overall I still maintain that there are some fundamental issues with the paper which make me stick to my original ratings.

---

> > > ### Author Response · Authors · 2018-11-22
> > > **Response to the comment “Still not convinced”**
> > >
> > > We appreciate your insightful comments.
> > >
> > > We agree with the fact that the words in a generated response are not independent indeed. Nevertheless, this assumption is somehow the very basis of quite a number of tasks such as machine translation and generative chat-bot.
> > > Actually, this assumption is equivalent to the Unigram Language Model, which usually takes an essential role implicitly/explicitly in many studies. Taking Hashimoto et al. (2018) as an example, all the words possibly appearing in the target sentence are directly predicted together via a multi-label classification layer (see Section 3.2, Vocabulary Prediction as Multi-Label Classification). It should be noted that, without the independence assumption, the multi-label classification strategy can not make sense.
> > > Hashimoto et al. (2018) have utilized the independence assumption explicitly, indeed. However, in the studies of Weng et al. (2017) and Wu et al. (2018), this assumption is taken as a fundamental common sense in the target sentence generation phase, and thus we can get obvious evidence in Equation 14 of (Weng et al., 2017) and Equation 10 of (Wu et al., 2018).
> > >
> > > Thanks for pointing out the issues in Equation 9 and 11. In fact, some fault has been made in the modification of Equation 9 in the previously updated version. Thus, it does need a sign-swapping in the max term.
> > > We have modified Equation 9 and 11 in the revised version.

---

### Public Comment · (anonymous) · 2018-10-25
**Nice work and some questions**

Thank you for your good work on open domain conversation generation. This paper provides a detailed analysis of the safe response problem from a mathematical view. Prior works try to explain this problem intuitively (one-to-many phenomenon), but this paper gives a new explanation with probability theory. However, I cannot understand some equations clearly.

(1) How to get the last part of Equation 8. Could you give more details about this equation?
(2) Why not choose a general response as a negative one for Equation 9?
(3) Why the gradient (Eq. 10) can be rewritten as Eq. 11.

---

> ### Author Response · Authors · 2018-10-26
> **Reply to "Nice work and some questions"**
>
> Thank you for your attention to our work.
>
> Question (1).
> Firstly, please accept our apologies for miswriting the K in the last part as 1/K, yet the conclusion that K * \epsilon is also proportional to \epsilon remains the same. To make it clear, according to Lemma 4, the middle and the right parts of Equation 8 can be derived as follows:
>
> log p(y|S(y)) + log ( p(x|y) / (p(x|y)p(y|S(y) + \epsilon))
> = log ( p(y|S(y))p(x|y) / (p(x|y)p(y|S(y) + \epsilon) )
> = log ( p(y|S(y)) / (p(y|S(y)) + \epsilon / p(x|y)) )
> \propto log ( p(y|S(y)) / (p(y|S(y)) + K \epsilon) )
>
> We’ve fixed this issue in our draft.
>
> Question (2).
> It’s a good question.
>
> According to our analysis, NRG models prefer universal replies due to the special properties in word and sentence level distributions.
>
> From the sentence level perspective, a straight-forward method of restraining the safe responses is to remove those high-frequent responses. However, simply removing these responses often leads to a mediocre experimental result, since there are always other high-frequent responses left in the remaining corpus (“high-frequent” is a comparative concept). Also, it is hard to capture all universal replies due to the variety of universal patterns, prefixes and suffixes existed inside the replies. Choosing general responses as negative samples is similar to above removing strategy, which might fail to punish all the universal replies.
>
> On the other hand, our objective is to enhance the importance of meaningful words during model training rather than punishing those high-frequent words, since high-frequent words themselves are important for sentence coherence of replies. For this purpose, we choose to use the random sampling method since it will not change the overall word distribution (especially the distribution of high-frequent words) in the negative samples.
>
> Question (3).
> The Eq. 10 is not equal to Eq. 11, they are the gradient functions for Eq. 9 corresponding to the two different conditions in the “max” part of Eq. 9.
>
> The Eq. 10 stands for the gradient when log p(y|x) – log p(y^-|x) <= \gamma, and the Eq. 11 stands for the opposite case.

---

### Author Response · Authors · 2018-10-30
**A non-direct derivation in Section 2.2 due to the paper version negligence.**

Dear reviewers and readers,

We have noticed that in the post-submission, it states that the Eq.2 is derived directly using Jensen’s Inequality. In this post, we would like to clarify that though Eq.2 still holds, in fact, it is indirectly derived using Jensen’s Inequality.

In more details, the size of \cup_k^K{S(y_k)} noted as L_S should be included in Equation 2 when performing Jensen’s Inequality. And L_S <= K*T, where K is the number of responses of x and T is the fixed number of words in a response. Then the last inequality in Equation 2 should be derived as follows.
\sum \log p(w|x) <= L_S \log \sum[p(w|x) / L_S]
To simplify, we use \sum to denote \sum_{}^{}.

Then, the Eq. 3 analyzes the last term in the previous inequality:
L_{S} \log \sum_{w \in \cup_k^K S(y_k)} \frac{p(w|x)}{L_{S}}
= L_{S} \log \frac{E(w|x) * T}{K*T*L_{S}}
\propto \log \frac{1}{(K*L_{S})^{L_{S}}}
\leq \frac{1}{K^{2K}}
which is much smaller than 1 / K, so that the analysis still holds.

We will update the manuscript to the right version when the rebuttal and discussion period opens.

Sincerely yours.

Authors

---

### Meta-Review · Area_Chair1 · 2018-12-17
**Important question, but unconvincing treatment here**

**Confidence:** 5
**Recommendation:** Reject

**Metareview:**

This paper seeks to shed light on why seq2seq models favor generic replies. The problem is an important one, unfortunately the responses proposed in the paper are not satisfactory. Most reviewers note problems and general lack of rigorousness in the assumptions used to produce the theoretical part of the paper (e.g., strong assumption of independence of generated words). The experiments themselves are not convincing enough to warrant acceptance by themselves.

---

> ### Public Comment · ~Sam_Brooks1 · 2019-06-08
> **Models**
>
> The existing Seq2Seq models are prone to producing short and generic replies, which blocks such neural network architectures from being utilized in practical open-domain response generation tasks.
> https://www.geometrydash.me/